# SEGMENT ANY 3D OBJECT WITH LANGUAGE

**Seungjun Lee**[∗], **Yuyang Zhao**,[∗] **Gim Hee Lee**
Department of Computer Science, National University of Singapore
{seungjun.lee, yuyang.zhao, gimhee.lee}@comp.nus.edu.sg
https://cvrp-sole.github.io

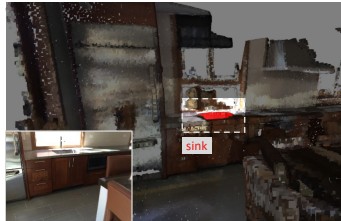 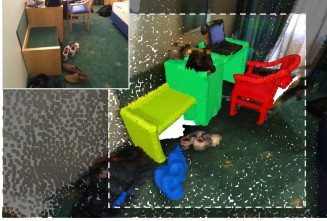 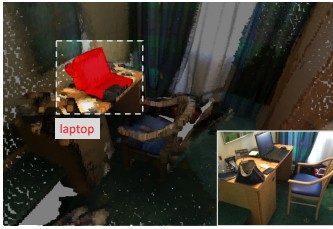

(a) *"Can I wash my hands?"*    (b) *"Brown Furnitures."*    (c) *"Device to play game."*

Figure 1: **Qualitative results of SOLE with various language instructions.** SOLE is highly generalizable and can segment corresponding instances with various language instructions, including but not limited to (a) *visual questions*, (b) *attributes description*, and (c) *functional description*.

## ABSTRACT

In this paper, we investigate Open-Vocabulary 3D Instance Segmentation (OV-3DIS) with free-form language instructions. Earlier works mainly rely on annotated base categories for training which leads to limited generalization to unseen novel categories. To mitigate the poor generalizability to novel categories, recent works generate class-agnostic masks or projecting generalized masks from 2D to 3D, subsequently classifying them with the assistance of 2D foundation model. However, these works often disregard semantic information in the mask generation, leading to sub-optimal performance. Instead, generating generalizable but semantic-aware masks directly from 3D point clouds would result in superior outcomes. To the end, we introduce Segment any 3D Object with LanguagE (**SOLE**), which is a semantic and geometric-aware visual-language learning framework with strong generalizability by generating semantic-related masks directly from 3D point clouds. Specifically, we propose a multimodal fusion network to incorporate multimodal semantics in both backbone and decoder. In addition, to align the 3D segmentation model with various language instructions and enhance the mask quality, we introduce three types of multimodal associations as supervision. Our SOLE outperforms previous methods by a large margin on ScanNetv2, ScanNet200, and Replica benchmarks, and the results are even closed to the fully-supervised counterpart despite the absence of class annotations in the training. Furthermore, extensive qualitative results demonstrate the versatility of our SOLE to language instructions.

## 1 INTRODUCTION

3D instance segmentation which aims at detecting, segmenting and recognizing object instances in 3D scenes is one of the crucial tasks for 3D scene understanding. Effective and generalizable 3D instance segmentation has great potential in real-world applications, including but not limited to autonomous driving, augmented reality (AR), and virtual reality (VR). Owing to its significance, 3D instance segmentation has achieved remarkable success in the computer vision community (Schult et al., 2022; Vu et al., 2022; He et al., 2022). Previous 3D instance segmentation models mainly focus on the closed-set setting, where the training and testing stages share the same categories. However, novel and unseen categories with various shapes and semantic meaning are inevitable in real-world applications. Failure to segment such instances drastically narrows the scope of application.

---

[∗]Equal Contribution

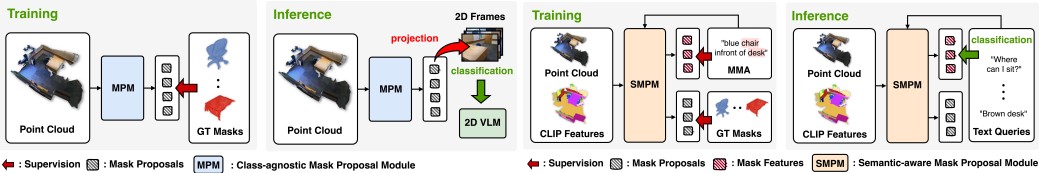

(a) Class agnostic 3D masks and classification      (b) Semantic-aware 3D masks (ours)

Figure 2: **Left (a) :** Previous works train class-agnostic mask proposal module with only using mask annotations. In the inference time, generated 3D masks are projected to 2D images and subsequently classified with the help of 2D foundation model. **Right (b) :** In contrast, we train semantic-aware mask proposal module with giving Multi-Modal Associations (MMA) and mask annotations as supervision. With regarding both geometry and semantic information in the mask generation, our SOLE can produce highly generalizable segments.

In view of the strong limitations of closed-set setting, open-set 3D instance segmentation (OS-3DIS) that aims at detecting and segmenting unseen classes based on instructions is introduced and investigated in the community. Most of the works (Huang et al., 2023b; Ding et al., 2022b; Nguyen et al., 2024) leverage category names or descriptions as segmentation instructions, which is also termed as open-vocabulary 3D instance segmentation (OV-3DIS). The early approaches (Ding et al., 2022b; Yang et al., 2023; Ding et al., 2023) split categories in each dataset into *base* and *novel* set. Only base categories are available in the training stage, but the model is expected to segment novel categories during inference. Due to the lack of novel classes during training, these methods easily overfit to the base categories and thus yielding sub-optimal performance on novel categories. In addition, they suffer from severe performance degradation when they are evaluated on the data with different distributions. In this regards, recent works (Takmaz et al., 2023; Huang et al., 2023b; Yan et al., 2024; Lu et al., 2023) explore more generalizable OV-3DIS with the help of 2D foundation models (Radford et al., 2021; Oquab et al., 2023; Zhou et al., 2022b). Specifically, Nguyen et al. (2024); Takmaz et al. (2023); Huang et al. (2023b) learn class-agnostic 3D masks from mask annotation and then project the point clouds to 2D images to obtain class labels from foundation models (Fig. 2a). Yan et al. (2024); Lu et al. (2023); Yin et al. (2024) predict 2D instances with 2D open-vocabulary instance segmentation model (Zhou et al., 2022b) and fuse them to obtain 3D predictions. However, class-agnostic masks and 2D projected masks ignore the semantic and geometry information in the mask generation, respectively, leading to the sub-optimal performance (details in Appendix. E.1). To this end, we investigate the OV-3DIS problem from the perspective of mask generator in this paper (Fig. 2b). We propose the semantic-aware mask generator to obtain semantic-related masks from 3D point clouds, yielding better and more generalizable 3D masks.

In this paper, we propose **SOLE**: Segment any 3D Object with LanguagE to circumvent the above-mentioned issues for OV-3DIS. To realize generalizable open-set 3D instance segmentation, our SOLE requires two main attributes: generating and classifying 3D masks directly from 3D point clouds, and responsive to free-form language instructions. The 3D segmentation network is required to be aligned with language instructions to directly segment and classify instances from point clouds. To this end, we build a multimodal fusion network with two main techniques: 1) Point-wise CLIP features obtained from pre-trained multimodal 2D segmentation model (Ghiasi et al., 2022) are incorporated to the backbone to enhance the generality of the model; 2) Cross-modality decoder is introduced to integrate information from language-domain features, facilitating the effective fusion of multimodal knowledge. Furthermore, we improve the generalization ability across various scene and language instructions with a novel visual-language learning framework, training the 3D segmentation network with three types of multimodal associations: 1) *Mask-visual association* aligns the 3D visual information with 2D visual information in the foundation model; 2) *mask-caption association* aims at enabling the mask generation and classification from various language forms; and 3) *mask-entity association* introduces fine-grained textual information into the framework, enhancing the ability of instance segmentation. These associations improve the language instruction alignment and enhance the 3D mask prediction with more abundant semantic information.

Equipped with a multimodal fusion network and three types of multimodal associations, our visual-language learning framework (SOLE) outperforms previous works by a large margin on ScanNetv2 (Dai et al., 2017), ScanNet200 (Rozenberszki et al., 2022) and Replica (Straub et al., 2019) benchmarks. Furthermore, SOLE can respond to free-form queries, including but not limited to

questions, attributes description, and functional description (Fig. 1 and Fig. 7). In summary, our contributions are as follows:

- We propose a visual-language learning framework for OV-3DIS, SOLE. A multimodal fusion network is designed for SOLE, which can directly predict semantic-related masks from 3D point clouds with multimodal information, leading to high-quality and generalizable segments.
- We propose three types of multimodal associations to improve the alignment between 3D segmentation model with the language. The associations improve the mask quality and the response ability to language instructions.
- SOLE achieves state-of-the-art results on ScanNetv2, Scannet200 and Replica benchmarks, and the results are even close to the fully-supervised counterpart. In addition, extensive qualitative results demonstrate that SOLE can respond to various language questions and instructions.

## 2 RELATED WORK

**Closed-Set 3D Instance Segmentation.** 3D instance segmentation aims at detecting, segmenting and recognizing the object instances in 3D scenes. Previous works (He et al., 2021; 2022; Ngo et al., 2023; Schult et al., 2022; Sun et al., 2023; Vu et al., 2022; Yi et al., 2019; Zhang et al., 2021; Hou et al., 2019; Yang et al., 2019a; Chen et al., 2021; Dong et al., 2022; Jiang et al., 2020; Liu et al., 2022; Wu et al., 2022) mainly consider the closed-set setting, where the training and testing categories are the same. These methods vary in feature extraction and decoding process. With the development of transformer models, mask prediction becomes a more efficient and effective way than traditional box detection decoding approaches. Mask3D (Schult et al., 2022) samples a fixed number of points across the scene as queries, and then directly predicts the final masks with attention mechanism, achieving better results. However, closed-set methods lack the capability to handle the unseen categories and thus hindering their application in the real world.

**Open-Vocabulary 2D Segmentation.** Owing to the recent success of large-scale vision-language models (Alayrac et al., 2022; Cherti et al., 2023; Girdhar et al., 2023; Jia et al., 2021; Radford et al., 2021; Yu et al., 2022; Yuan et al., 2021), notable achievements have been made in open-vocabulary or zero-shot 2D segmentation (Ding et al., 2022a; Ghiasi et al., 2022; Gu et al., 2021; He et al., 2023; Kuo et al., 2022; Li et al., 2022; Liang et al., 2023; Ma et al., 2022; Rao et al., 2022; Xu et al., 2022; 2023; Zabari & Hoshen, 2021; Zhou et al., 2022a; Cho et al., 2023; Ma et al., 2023; Yuan et al., 2024). The common key idea is to leverage 2D mulitmodal foundation models (Radford et al., 2021; Jia et al., 2021) for the transfer of image-level embeddings to the pixel-level downstream tasks. LSeg (Li et al., 2022), OpenSeg (Ghiasi et al., 2022), and OVSeg (Liang et al., 2023) align pixel-level or mask-level visual features to text features from foundation model for open-vocabulary semantic segmentation. Recently, Open-Vocabulary SAM (Yuan et al., 2024) integrates CLIP and SAM, further exploring interactive open-world tasks. Other works such as X-Decoder (Zou et al., 2023), FreeSeg (Qin et al., 2023) and SEEM (Zou et al., 2024) suggest more unified-framework for open-vocabulary segmentation, include instance, panoptic, and referring segmentation.

**Open-Vocabulary 3D Scene Understanding.** The remarkable success achieved in open-vocabulary 2D segmentation (OV-2DS) has spurred several endeavors in open-vocabulary 3D segmentation. However, the techniques in OV-2DS cannot be directly transferred to the 3D domain due to the lack of 3D multimodal foundation model. Consequently, researchers propose to align 2D images and 3D point clouds and thus lifting 2D foundation models to 3D. For open-vocabulary 3D semantic segmentation, (Chen et al., 2023; Ding et al., 2022b; Ha & Song, 2022; Huang et al., 2023a; Jatavallabhula et al., 2023; Peng et al., 2023; Shafiullah et al., 2022; Shah et al., 2023) construct task-agnostic point-wise feature representations from 2D foundation models (Radford et al., 2021), and then use these features to query the open-vocabulary concepts within 3D scene. These works focus purely on transferring semantic information from 2D to 3D, limiting the application for instance-level recognition tasks. In this regard, open-vocabulary 3D instance segmentation (OV-3DIS) (Ding et al., 2023; Huang et al., 2023b; Takmaz et al., 2023; Lu et al., 2023; Nguyen et al., 2024; Yan et al., 2024) is introduced to detect and segment instances of various categories in 3D scenes. PLA (Ding et al., 2022b) and its variants (Yang et al., 2023; Ding et al., 2023) split the training categories into base and novel classes, and train the model only with base class annotation. OpenMask3D (Takmaz et al., 2023) and OpenIns3D (Huang et al., 2023b) learn class-agnostic 3D masks from mask annotations and then use the corresponding 2D images to obtain class labels from foundation models. With

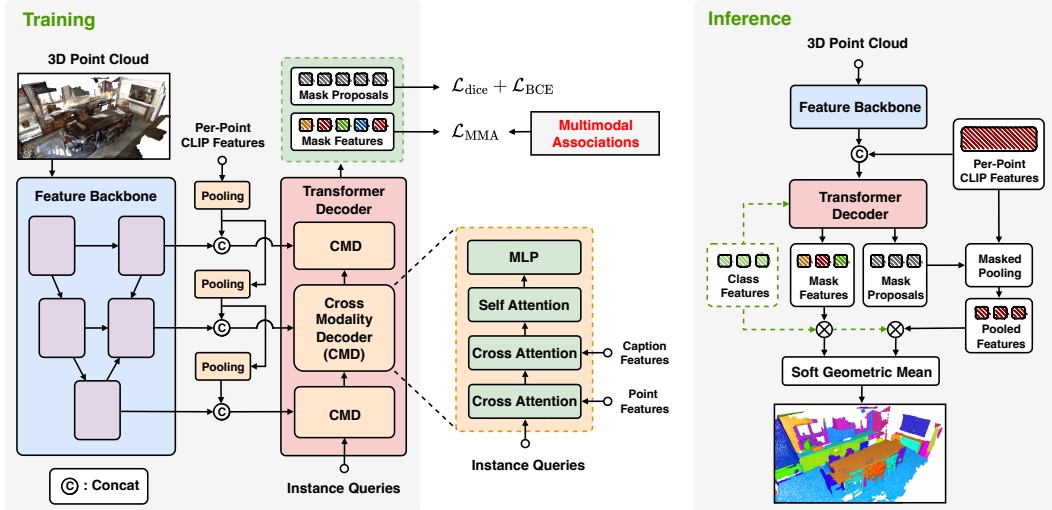

Figure 3: **Overall framework of SOLE.** SOLE is built on transformer-based instance segmentation model with multimodal adaptations. For model architecture, backbone features are integrated with per-point CLIP features and subsequently fed into the cross-modality decoder (CMD). CMD aggregates the point-wise features and textual features into the instance queries, finally segmenting the instances, which are supervised by multimodal associations. During inference, predicted mask features are combined with the per-point CLIP features, enhancing the open-vocabulary performance.

following similar paradigm, Open3DIS (Nguyen et al., 2024) further improves the quality of masks by fusing 3D segments and 2D masks. Recently, researchers also investigate direct lifting of 2D predictions from 2D instance segmentation model (Zhou et al., 2022b) to 3D without training (Yan et al., 2024; Lu et al., 2023; Yin et al., 2024). However, previous works often disregard semantic information in the mask generation, leading to the poor semantic generalization ability. Considering the limitations of previous works, we significantly improve OV-3DIS by designing a visual-language learning framework with a multimodal network and various multimodal associations.

## 3 METHOD

**Objective.** The goal of open-vocabulary 3D instance segmentation (OV-3DIS) with free-form language instructions is defined as follows: Given a 3D point cloud $\mathbf{P} \in \mathbb{R}^{M \times C}$, the corresponding 2D images $I$ and the instance-level 3D masks $\mathbf{m}$, we aim to train a 3D instance segmentation network without ground-truth class annotations. During inference, given a text prompt $q$, the trained 3D instance segmentation network must detect and segment corresponding instances.

**Mask-Prediction Baseline.** We build our framework on the transformer-based 3D instance segmentation model Mask3D (Schult et al., 2022), which treats the instance segmentation task as the mask prediction paradigm. Specifically, the transformer decoders with mask queries are used to segment instances. Given $N_q$ queries selected from the scene, cross attention is used to aggregate information from the point clouds to instance queries. After several decoder layers, $N_q$ queries become $N_q$ masks with corresponding semantic prediction. During training, Hungarian matching (Kuhn, 1955) is adopted to match and train the model with ground truth labels and masks. At the inference stage, $N_q$ masks with correct semantic classification results are taken as the final outputs. Our SOLE leverages the mask prediction paradigm with transformer-based architecture, where the model is only trained with masks without ground truth labels to achieve generalizable OV-3DIS.

**Overview.** The overall architecture of SOLE is illustrated in Fig. 3. To realize open-vocabulary instance segmentation with free-form language instructions, we improve the transformer-based instance segmentation model with multimodal information: point-wise CLIP features in the backbone (Sec. 3.1) and textual information in the decoder (Sec. 3.2). Furthermore, to achieve better generalization ability without ground truth class labels, we construct three types of multimodal associations on target instances: mask-visual association, mask-caption association and mask-entity association

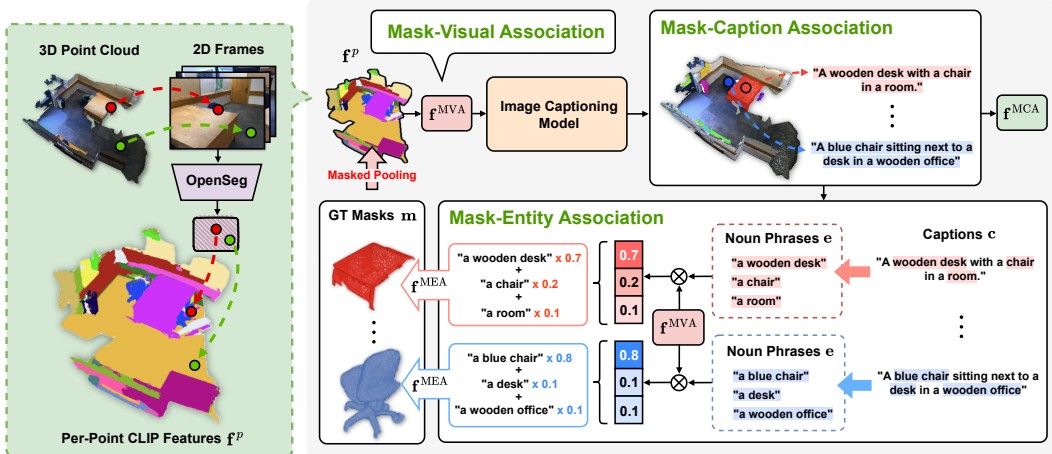

Figure 4: **Three types of multimodal association instance.** For each ground truth instance mask, we first pool the per-point CLIP features to obtain *Mask-Visual Association* $\mathbf{f}^{\text{MVA}}$. Subsequently, $\mathbf{f}^{\text{MVA}}$ is fed into CLIP space captioning model to generate caption and corresponding textual feature $\mathbf{f}^{\text{MCA}}$ for each mask, termed as *Mask-Caption Association*. Finally, noun phrases are extracted from mask caption and the embeddings of them are aggregated via multimodal attention to get *Mask-Entity Association* $\mathbf{f}^{\text{MEA}}$. The three multimodal associations are used for supervising SOLE to acquire the ability to segment 3D objects with free-form language instructions.

to train SOLE. Equipped with the multimodal framework and associations, our SOLE can effectively segment instances given various language prompts.

## 3.1 BACKBONE FEATURE ENSEMBLE

Initializing the backbone with pre-trained model (Jia et al., 2022; Zhao et al., 2023b;a) is an efficient and effective way to improve the performance on the downstream tasks, especially when the downstream data is not in abundance. For 3D open-set setting, leveraging 2D foundation model is crucial due to the limited 3D data. We thus follow (Peng et al., 2023) to project pre-trained visual features of 2D images to 3D point clouds based on the camera pose. To maintain the fine-grained and generalizable features, we leverage OpenSeg (Ghiasi et al., 2022) as the 2D backbone. These features contain visual information in the CLIP (Radford et al., 2021) feature space, which is aligned with textual information.

Since CLIP feature space mainly focuses on semantic information due to the image-level contrastive training, leveraging the projected features solely cannot achieve optimal performance on instance segmentation. To this end, we train a 3D instance segmentation backbone and combine its features $\mathbf{f}^b \in \mathbb{R}^{M \times D}$ with the projected 2D CLIP features $\mathbf{f}^p \in \mathbb{R}^{M \times C}$:

$$\tilde{\mathbf{f}}^b = \text{concat}(\mathbf{f}^p, \mathbf{f}^b) \in \mathbb{R}^{M \times (C+D)}, \tag{1}$$

where $M$ denotes the number of points while $D$ and $C$ denote the feature dimension of 3D instance segmentation backbone and the projected 2D features, respectively. Note that features of different resolutions are extracted from the 3D backbone and respectively incorporated with the 2D CLIP features. As illustrated in Fig. 3, the same pooling strategy with 3D backbone is adopted to CLIP features, aligning the resolution. Finally, incorporated point-wise features with multiple resolutions are fed into cross modality decoder.

## 3.2 CROSS MODALITY DECODER (CMD)

Projected 2D CLIP features provide generalizable visual information but the language information is not explicitly integrated, limiting the responsive ability to language instructions. To circumvent this issue, we introduce Cross Modality Decoder (CMD) to incorporate textual information in the decoding process of our framework. Specifically, each CMD module contains three attention layers. Instance queries first extract visual information from the CLIP-combined backbone features $\tilde{\mathbf{f}}^b$.

CLIP textual features are then projected to key and value in the second attention layer, incorporating the text domain knowledge. During the training, CLIP textual features are obtained from the caption features of each target mask, $\mathbf{f}^{\text{MCA}} \in \mathbb{R}^{N_c \times C}$ (See Sec. 3.3 for details), whereas, during the inference, it can be the description of the query instance or other form of language instructions, such as visual questions or functional attributes. Finally, self-attention is applied to the instance queries to further improve the representation. By fusing the multimodal knowledge from CLIP with multi-level CMD as the decoder, SOLE can respond to various language instructions with high-quality results.

## 3.3 Vision-Language Learning

We do vision-language learning to enable our SOLE towards generalizable OV-3DIS. To respond effectively to various language instructions, we leverage multimodal information stemming from target mask annotations, to supervise the segmentation network. Specifically, three types of supervision in hierarchical granularity are proposed: 1) mask-visual association, 2) mask-caption association and 3) mask-entity association.

**Mask-Visual Association (MVA).** Using the correspondence between 2D images and 3D point clouds, we can get the instance-level CLIP visual features $\mathbf{f}^{\text{MVA}} \in \mathbb{R}^{N_m \times C}$ by averaging the per-point CLIP features within the $N_m$ target instance masks $\mathbf{m} = [m_1, m_2, \ldots, m_{N_m}]$. The instance-level CLIP visual features can serve as the supervision to indirectly align the 3D segmentation model to CLIP textual space. In addition, as the intermediate representation between 3D point cloud and language, the mask-visual association is also the basis for the following two fine-grained associations.

**Mask-Caption Association (MCA).** Despite being in the CLIP feature space, mask-visual association is not an accurate and precision language supervision. Instead, directly supervising the model with language instructions would yield better results. Due to the strong generalization ability of CLIP (Radford et al., 2021), text generation from CLIP space is widely investigated in the community (Tewel et al., 2022; Mokady et al., 2021; Li et al., 2023). Since the instance-level CLIP visual features $\mathbf{f}^{\text{MVA}}$ in the mask-visual association is in CLIP visual space, we can feed them to the CLIP space caption generation model (DeCap (Li et al., 2023)) to obtain the mask captions $\mathbf{c} = [c_1, c_2, ..., c_{N_m}]$. The mask captions are then fed into CLIP textual model to extract the mask-caption association $\mathbf{f}^{\text{MCA}}$. This association represents the language information for the instance masks, used in CMD to fuse textual information during the training.

**Mask-Entity Association (MEA).** Although mask-caption association can provide detailed language descriptions for both semantics and geometry, it may be ambiguous for specific categories. As shown in the example of Fig. 4. The mask caption for a desk is "A wooden desk with a chair in a room". Such caption can lead to the confusion of the model between the chair and the desk, or misinterpretation of the two instances as a single one. It is therefore important to introduce a more fine-grained visual-language association for better semantic learning.

Since the objects are commonly the nouns in the caption, we can extract the entity-level descriptions for the nouns and match them with the instances. Specifically, as illustrated in Fig. 4, we first extract all the noun phrases $\mathbf{e}_i$ for each mask caption $c_i$ and obtain the text feature of each noun phrase from CLIP text encoder $\mathcal{T}$ as below:

$$\mathcal{E}(c_i) = \mathbf{e}_i = [e_1, e_2, \ldots, e_{N_e^i}], \quad \mathbf{f}_i^{\mathbf{e}} = \mathcal{T}(\mathbf{e}_i) \in \mathbb{R}^{N_e^i \times C}, \tag{2}$$

where $\mathcal{E}(\cdot)$ denotes the NLP tool to extract noun phrases and $N_e^i$ denotes the number of nouns obtained from mask caption $c_i$. The entities can be matched to the mask in either a hard or soft manner. Intuitively, the most similar entity can be viewed as the mask label. However, there are two main issues with such a hard matching. First, the generated caption and the similarity results may not be accurate, leading to wrong supervision. Second, although the entity is correct, hard matching ignores the geometry information in the context and thus impairing the responsive ability to language instructions. To this end, we propose a soft matching to get mask-entity association by multimodal attention. Specifically, the aggregated entity feature for the $i$-th mask $\hat{\mathbf{f}}_i^{\text{MEA}}$ is obtained based on the attention map $A_{c,e}$ constructed by cosine similarity $\cos(\cdot, \cdot)$ between mask feature and

entity features:

$$\mathbf{f}_i^{\text{MEA}} = A_{c,e} \cdot \mathbf{f}_i^{\mathbf{e}} = \sum_k^{N_e^i} \frac{\exp\left(\cos\left(\mathbf{f}_i^{\text{MVA}}, \mathbf{f}_i^{e_k}\right)\right)}{\sum_j^{N_e^i} \exp\left(\cos\left(\mathbf{f}_i^{\text{MVA}}, \mathbf{f}_i^{e_j}\right)\right)} \cdot \mathbf{f}_i^{e_k}, \tag{3}$$

where $\mathbf{f}_i^{\text{MVA}}$ denotes the mask-visual association feature for $i$-th mask, and $\mathbf{f}_i^{e_k}$ is the CLIP textual feature for $k$-th entity in the $i$-th mask caption. With the aggregated entity feature, the 3D mask can be aligned with a specific instance category.

### 3.4 TRAINING AND INFERENCE

**Training.** The three types of multimodal associations are effective supervision to learn a generalizable 3D instance segmentation model. We follow the mask prediction paradigm to train the segmentation model, which matches the ground truth instances with the predicted masks via Hungarian matching (Kuhn, 1955). Specifically, the matching cost between $i$-th predicted mask and $j$-th ground truth instance is calculated as:

$$\begin{aligned} \mathcal{C}(i,j) = &- \lambda_{\text{MMA}} \left( p\left(\cos\left(\mathbf{f}_i^{\mathbf{m}}, \mathbf{f}_j^{\text{MVA}}\right)\right) + p\left(\cos\left(\mathbf{f}_i^{\mathbf{m}}, \mathbf{f}_j^{\text{MCA}}\right)\right) + p\left(\cos\left(\mathbf{f}_i^{\mathbf{m}}, \mathbf{f}_j^{\text{MEA}}\right)\right) \right) \\ &+ \lambda_{\text{dice}}\mathcal{L}_{\text{dice}}(i,j) + \lambda_{\text{BCE}}\mathcal{L}_{\text{BCE}}(i,j), \end{aligned} \tag{4}$$

where $p(\cdot)$ denotes the softmax probability following the cosine similarity between the predicted instance and the ground truth. After matching the masks and ground truth instances, the model is trained with the combination of mask and semantic loss.

Specifically, all three types of associations are used to semantically supervise the model. For each association, we follow (Zhou et al., 2023) to use the combination of focal loss (Lin et al., 2017) and dice loss, which can ensure the segmentation result for each class is independently generated. The semantic multimodal association loss $\mathcal{L}_{\text{MMA}}^j$ for $j$-th ground truth mask is:

$$\mathcal{L}_{\text{MMA}}^j = \sum_a \left( \mathcal{L}_{\text{focal}}(\hat{p}_{\sigma(j)}^a, y_j^a) + \mathcal{L}_{\text{dice}}(\hat{p}_{\sigma(j)}^a, y_j^a) \right), \tag{5}$$

where $a \in \{\text{MVA}, \text{MCA}, \text{MEA}\}$ denotes three types of associations and $y_j^a$ is the binary label for matching. $\hat{p}_{\sigma(j)}^a = \text{sigmoid}(\mathbf{f}_{\sigma(j)}^{\mathbf{m}} \cdot \mathbf{f}_j^a)$ is the semantic probability between the prediction with the association $a$. The overall training loss is the combination of mask loss and semantic loss:

$$\mathcal{L} = \frac{1}{N_m} \sum_j^{N_m} \left( \lambda_{\text{MMA}}\mathcal{L}_{\text{MMA}}^j + \lambda_{\text{dice}}\mathcal{L}_{\text{dice}}(\hat{m}_{\sigma(j)}, m_j) + \lambda_{\text{BCE}}\mathcal{L}_{\text{BCE}}(\hat{m}_{\sigma(j)}, m_j) \right), \tag{6}$$

where $\hat{m}_{\sigma(j)}$ denotes matched predicted mask with $j$-th target mask.

**Inference.** During inference, we combine the visual feature from CLIP with the predicted mask feature to achieve better generalization ability. Specifically, after obtaining the 3D masks, per-point CLIP features are pooled within the mask. The pooled CLIP feature and mask feature are then given to the text classifier $\mathbf{f}^t$ to obtain the respective classification probability $\mathbf{P}^m = p(\cos(\mathbf{f}^m, \mathbf{f}^t))$ and $\mathbf{P}^p = p(\cos(\mathbf{f}^p, \mathbf{f}^t))$, and the final probability is yielded by soft geometric mean between them:

$$\mathbf{P} = \max\left(\mathbf{P}^m, \mathbf{P}^p\right)^\tau \cdot \min\left(\mathbf{P}^m, \mathbf{P}^p\right)^{1-\tau}, \tag{7}$$

where $\tau$ is the exponent to increase confidence, which we set to 0.667 in this paper. For benchmark evaluation, we use CLIP textual features of all category names as the classifier. For responding to other language instructions, we use the CLIP textual feature of corresponding language instruction as binary classifier.

## 4 EXPERIMENTS

### 4.1 EXPERIMENTAL SETTING

**Datasets.** We evaluate SOLE on the popular scene understanding datasets: ScanNetv2 (Dai et al., 2017), ScanNet200 (Rozenberszki et al., 2022) and Replica (Straub et al., 2019) in both closed-set and open-set 3D instance segmentation tasks. ScanNetv2 (Dai et al., 2017) is a popular indoor

Table 1: **The comparison of closed-set 3D instance segmentation setting on ScanNetv2.** SOLE is compared with mask-training methods and the fully-supervised counterpart (upper bound). SOLE outperforms all the OV-3DIS methods and achieves competitive results with the fully-supervised model. The best results are in **bold** while the second best results are underscored.

| Method | AP | AP$_{50}$ | AP$_{25}$ | voxel size |
|---|---|---|---|---|
| OpenIns3D (Huang et al., 2023b) | - | 28.7 | 38.9 | 2cm |
| OpenMask3D (Takmaz et al., 2023) | 31.0 | 39.5 | 44.0 | 2cm |
| Open3DIS (only 3D) (Nguyen et al., 2024) | 31.3 | 42.4 | 47.8 | 2cm |
| SOLE *w 4cm voxel size* | 30.8 | 52.5 | 70.9 | 4cm |
| SOLE *w/o text sup* | 35.0 | 50.2 | 60.2 | 2cm |
| **SOLE (*ours*)** | **44.4** (+13.1) | **62.2** (+19.8) | **71.4** (+23.6) | 2cm |
| Mask3D (Schult et al., 2022) (*fully sup*) | 55.2 | 73.7 | 83.5 | 2cm |

Table 2: **The comparison of closed-set 3D instance segmentation setting on ScanNet200.** SOLE is compared with mask training methods on the overall segmentation performance and on each subset. SOLE significantly outperforms state-of-the-art methods on five out of the six evaluation metrics under the same conditions using only proposals from a 3D Network.

| Method | AP | AP$_{50}$ | AP$_{25}$ | AP$_{head}$ | AP$_{com}$ | AP$_{tail}$ |
|---|---|---|---|---|---|---|
| OpenIns3D (Huang et al., 2023b) | 8.8 | 10.3 | 14.4 | 16.0 | 6.5 | 4.2 |
| OpenMask3D (Takmaz et al., 2023) | 15.4 | 19.9 | 23.1 | 17.1 | 14.1 | **14.9** |
| Open3DIS *(only 3D)* (Nguyen et al., 2024) | 18.6 | 23.1 | 27.3 | 24.7 | 16.9 | 13.3 |
| **SOLE (*ours*)** | **20.1** | **28.1** | **33.6** | **27.5** | **17.6** | 14.1 |
| | (+1.5) | (+5.0) | (+6.3) | (+2.8) | (+0.7) | (-0.8) |
| Mask3D (*fully sup*) (Schult et al., 2022) | 26.9 | 36.2 | 41.4 | 39.8 | 21.7 | 17.9 |

point cloud dataset with 18 instance classes, where "other furniture" class is disregarded due to its ambiguity. ScanNet200 (Rozenberszki et al., 2022) is a fine-grained annotated version of ScanNetv2 that contains 200 classes of head (66 categories), common (68 categories) and tail (66 categories) subsets. For ScanNetv2 and ScanNet200, we evaluate the closed-set setting and the hierarchical open-set setting. Replica (Straub et al., 2019) is a high-quality synthetic dataset annotated with 48 instance categories. Following (Takmaz et al., 2023), we evaluate on eight scenes in Replica for open-set instance segmentation, including {office0, office1, office2, office3, office4, room0, room1 and room2.}

**Implementation Details.** Following the Mask3D (Schult et al., 2022), we adopt Minkowski-UNet (Choy et al., 2019) as backbone. The feature backbone extracts point features in 5 scales, while 4 layers of transformer decoder iteratively refine the instance queries. Our model is trained for 600 epochs with AdamW (Loshchilov & Hutter, 2017) optimizer. The learning rate is set to $1 \times 10^{-4}$ with cyclical decay. In training, we set $\lambda_{MMA} = 20.0$, $\lambda_{dice} = 2.0$ and $\lambda_{BCE} = 5.0$ as the loss weight.

**Baselines.** We compare SOLE mainly with recent mask-training methods (Takmaz et al., 2023; Huang et al., 2023b; Nguyen et al., 2024). Mask-training methods (Takmaz et al., 2023; Huang et al., 2023b; Nguyen et al., 2024) train class-agnostic mask generator with mask annotations and get the semantic prediction with 2D foundation models. Evaluation is conducted under the same conditions using only proposals from a 3D Network on all the categories. The proposed semantic-aware mask generator can be combined with the classification techniques proposed by previous works, further improving the performance (Appendix. D.2).

**Evaluation Metric.** Average precision (AP) of different IoU thresholds is adopted as the evaluation metric, including AP under 25%, 50% IoU and the average AP from 50% to 95% IoU.

## 4.2 COMPARISON WITH PREVIOUS METHODS

**Closed-Set 3D Instance Segmentation.** We compare our SOLE with mask-training methods (Takmaz et al., 2023; Huang et al., 2023b; Nguyen et al., 2024) on the closed-set 3D instance segmenta-

Table 3: **The comparison of hierarchical open-set 3D instance segmentation setting on Scan-Netv2 (Dai et al., 2017)→ScanNet200 (Rozenberszki et al., 2022).** SOLE is compared with Open-Mask3D and Open3DIS on both base and novel classes and achieves the best results.

| Method | Novel Classes | | | Base Classes | | | All Classes | |
|---|---|---|---|---|---|---|---|---|
| | AP | $AP_{50}$ | $AP_{25}$ | AP | $AP_{50}$ | $AP_{25}$ | AP | $AP_{tail}$ |
| OpenMask3D (Takmaz et al., 2023) | 11.9 | 15.2 | 17.8 | 14.3 | 18.3 | 21.2 | 12.6 | 11.5 |
| Open3DIS *(only 3D)* (Nguyen et al., 2024) | 14.9 | 19.2 | 22.1 | 16.5 | 22.4 | 26.2 | 15.3 | 10.9 |
| **SOLE (*ours*)** | **19.1** | **26.2** | **30.7** | **17.4** | **26.2** | **32.1** | **18.7** | **12.5** |
| | (+4.2) | (+7.0) | (+8.6) | (+0.9) | (+3.8) | (+5.9) | (+3.4) | (+1.0) |

Table 4: **The comparison of open-set 3D instance segmentation setting on ScanNet200 (Rozen-berszki et al., 2022)→Replica (Straub et al., 2019).** SOLE outperforms state-of-the-art methods (Takmaz et al., 2023; Nguyen et al., 2024) on all the evaluation metrics.

| Method | Mask Training | AP | $AP_{50}$ | $AP_{25}$ |
|---|---|---|---|---|
| OpenIns3D (Huang et al., 2023b) | ScanNet200 | 13.6 | 18.0 | 19.7 |
| OpenMask3D (Takmaz et al., 2023) | ScanNet200 | 13.1 | 18.4 | 24.2 |
| Open3DIS *(only 3D)* (Nguyen et al., 2024) | ScanNet200 | 14.9 | 18.8 | 23.6 |
| **SOLE (*ours*)** | ScanNet200 | **24.7** (+9.8) | **31.8** (+13.0) | **40.3** (+16.1) |

tion setting. From the comparison results in Tab. 1, we can make the following observations. **First**, with the same mask supervision, SOLE significantly surpasses state-of-the-art method (Nguyen et al., 2024) by 13.1%, 19.8%, and 23.6% on AP, $AP_{50}$ and $AP_{25}$, respectively. **Second**, our SOLE can even achieve competitive performance with the fully-supervised counterpart (44.4% *v.s.* 55.2% in AP) despite not using the class labels. Finally, we provide two variants of SOLE to further verify our effectiveness. SOLE *w 4cm voxel size* leverages 4cm voxel size instead of 2cm as in previous works. Larger voxel size can save the memory requirements and speed up the model with the loss of precision. Despite using a small voxel size, SOLE *w 4cm voxel size* is still on par with Open-Mask3D and Open3DIS in AP, and largely outperforms them in $AP_{50}$ and $AP_{25}$. Furthermore, we verify that the effectiveness of our framework is not limited to the caption model and NLP tools by conducting experiments without any additional textual information, *i.e.* SOLE *w/o text sup.* In this experiment, mask-caption association and mask-entity association is removed since the caption is not available. The model can still achieve the state-of-the-art performance despite only trained with mask-visual association. Additionally, we compare SOLE with mask-training methods on ScanNet200 (Rozenberszki et al., 2022) in Tab. 2. All of the methods are evaluated on the overall segmentation performance and the performance on each of the three subsets. SOLE outperforms state-of-the-art methods (Nguyen et al., 2024; Takmaz et al., 2023) on five out of the six metrics and achieves comparable performance on the tail classes. The results on ScanNet200 (Rozenberszki et al., 2022) further demonstrate the effectiveness of our framework.

**Hierarchical and Cross-Domain Open-Set 3DIS.** To evaluate the generalization capability of our work, we compare our SOLE with mask training methods (Huang et al., 2023b; Takmaz et al., 2023; Nguyen et al., 2024) in open-set setting, using Scannet200 (Rozenberszki et al., 2022) and Replica (Straub et al., 2019) datasets. For ScanNet200, both models are trained with mask annotations in ScanNetv2 (Dai et al., 2017). Following (Takmaz et al., 2023), 53 classes that are semantically close to the ScanNet, are grouped as "Base". The remaining 147 classes are grouped as "Novel". Both in-distribution ("base") and out-of-distribution ("novel") classes are reported in Tab. 3. Our SOLE outperforms OpenMask3D (Takmaz et al., 2023) and Open3DIS (Nguyen et al., 2024) by a large margin on both base and novel classes. Furthermore, to verify the generalization ability of SOLE when both domain shift and category shift exist, we compare our framework with all of the mask training methods on the synthetic Replica benchmark (Straub et al., 2019). Models are trained on the annotated masks on ScanNet200. As shown in Tab. 4, our method further shows superior robustness on more out-of-distribution data from Replica, achieving +9.8% improvement in AP score compared to Open3DIS.

### 4.3 ABLATION STUDIES AND ANALYSIS

In this section, we conduct several ablation studies to validate our design choices. All of the studies are evaluated on ScanNetv2 (Dai et al., 2017) dataset.

Table 5: **Multimodal fusion network.**

| No. | $\mathbf{f}^p$ | $\mathbf{f}^b$ | CMD | AP | AP$_{50}$ | AP$_{25}$ | voxel size |
|---|---|---|---|---|---|---|---|
| 1 | ✓ | | ✓ | 18.7 | 36.4 | 58.1 | 4cm |
| 2 | | ✓ | ✓ | 25.4 | 47.0 | 66.0 | 4cm |
| 3 | ✓ | ✓ | ✓ | **30.8** | **52.5** | **70.9** | 4cm |
| 4 | ✓ | ✓ | | 42.8 | 60.5 | 68.9 | 2cm |
| 5 | ✓ | ✓ | ✓ | **44.4** | **62.2** | **71.4** | 2cm |

Table 6: **Multimodal associations.**

| No. | $\mathbf{f}^{MVA}$ | $\mathbf{f}^{MCA}$ | $\mathbf{f}^{MEA}$ | AP | AP$_{50}$ | AP$_{25}$ | voxel size |
|---|---|---|---|---|---|---|---|
| 1 | ✓ | | | 24.5 | 42.0 | 56.0 | 4cm |
| 2 | | ✓ | | 30.4 | 53.0 | 68.7 | 4cm |
| 3 | | | ✓ | **32.1** | **53.8** | 70.0 | 4cm |
| 4 | ✓ | ✓ | | 29.1 | 50.9 | 66.8 | 4cm |
| 5 | | ✓ | ✓ | 30.3 | 53.7 | 70.4 | 4cm |
| 6 | ✓ | ✓ | ✓ | 30.8 | 52.5 | **70.9** | 4cm |

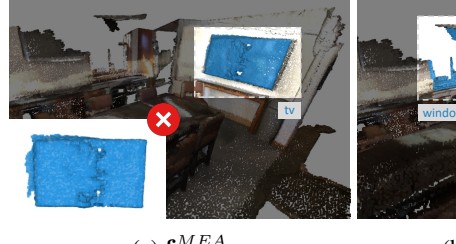
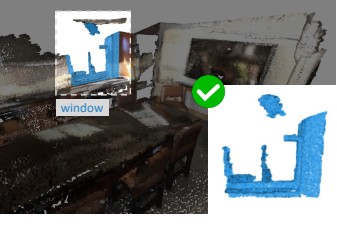
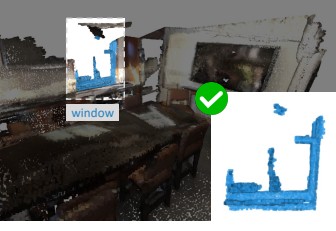

(a) $\mathbf{f}^{MEA}$      (b) $\mathbf{f}^{MVA}, \mathbf{f}^{MCA}$      (c) $\mathbf{f}^{MVA}, \mathbf{f}^{MCA}, \mathbf{f}^{MEA}$

Figure 5: **Qualitative analysis on multimodal associations.** Given the free-form language instruction, "*I wanna see outside.*", SOLE trained only with $\mathbf{f}^{\mathrm{MEA}}$ captures the wrong object ((a)), whereas it segments the related object when $\mathbf{f}^{\mathrm{MVA}}$ and $\mathbf{f}^{\mathrm{MCA}}$ are given as the supervision ((b) and (c)).

**Multimodel Fusion Network.** In Tab. 5, we conduct component analysis on multimodal fusion network, validating the effectiveness of backbone feature ensemble and Cross-Modality Decoder (CMD). As for the backbone feature ensemble, leveraging projected 2D CLIP features $\mathbf{f}^p$ (first row) as only backbone can have better semantic information but lack the 3D geometry detection ability, leading to poor semantic recognition ability. In contrast, solely using 3D instance backbone feature $\mathbf{f}^b$ (second row) cannot inherit the generalizable semantic information, resulting in sub-optimal performance. Combining the two features (third row) can make full use of generalized semantic information while learning good geometry detection ability from 3D masks, yielding optimal results. Additionally, Cross Modality Decoder (CMD) can further enhance the ability to understand language instructions, improving AP by 1.6%.

**Multimodal Associations.** We analyze the components of multimodal associations ($\mathbf{f}^{\mathrm{MVA}}$, $\mathbf{f}^{\mathrm{MCA}}$, and $\mathbf{f}^{\mathrm{MEA}}$) in Tab. 6, reporting the scores of various combinations on ScanNetv2 (Dai et al., 2017) with 4cm voxel size. We have the following observations. **First**, using any of multimodal associations can already achieve significant performance, outperforming previous state-of-the-art method (OpenIns3D (Huang et al., 2023b)) with larger voxel size (lower resolution). **Second**, among the three types of associations, mask-entity association $\mathbf{f}^{\mathrm{MEA}}$ is the most effective one on evaluation metrics since it can align the masks with specific categories. **Third**, when combining $\mathbf{f}^{\mathrm{MEA}}$ with the other two associations, the model suffers from performance degradation on AP and AP$_{50}$ while the performance improves on AP$_{25}$. This observation shows that mask-visual association and mask-caption association can help semantic learning but impair mask accuracy. To this end, we further illustrate qualitative results in Fig. 5. Given a free-form language instruction instead of category name, *e.g.*, "I wanna see outside", the model only using mask-entity association cannot segment the correct instance (Fig. 5a) while the model incorporating the other associations (Fig. 5b and Fig. 5c) can. Therefore, despite slightly impairing the performance on benchmark, mask-visual association and mask-caption association are crucial to recognizing free-form language instructions, benefiting the applications in real-world scenarios.

## 5   CONCLUSION

In this paper, we propose a novel framework, SOLE, for open-vocabulary 3D instance segmentation with free-form language instructions. SOLE contains a multimodal fusion network and is supervised with three types of multimodal associations, aiming at aligning the model with various free-form language instructions. Our framework outperforms previous methods by a large margin on three benchmarks while achieving competitive performance with the fully-supervised counterpart. Moreover, extensive qualitative results demonstrate the versatility of our SOLE to language instructions.

## ACKNOWLEDGEMENTS

This research work is supported by the Agency for Science, Technology and Research (A*STAR) under its MTC Programmatic Funds (Grant No. M23L7b0021), and the Tier 2 grant MOE-T2EP20124-0015 from the Singapore Ministry of Education.

## ETHICS STATEMENT

This research adheres to the ethical guidelines outlined in the ICLR Code of Ethics. In this section, we take following aspects into account for the ethical considerations.

**Data and Privacy :** The datasets used in this research are either publicly available (Straub et al., 2019) or accessible under the agreement on term of use (Dai et al., 2017; Rozenberszki et al., 2022).

**Potential Harmful Applications :** The proposed method for open-vocabulary 3D instance segmentation could be used in sensitive domains such as surveillance. We encourage future users to apply the model responsibly, with appropriate safeguards to mitigate potential misuse.

**Research Integrity :** We confirm that all experiments were conducted in accordance with best practices in research ethics, and results were reported with complete transparency and accuracy.

## REPRODUCIBILITY STATEMENT

To ensure the reproducibility of our work, we state specific implementation details in Sec. 4.1 and Appendix. Sec. A. The code will be made publicly available.

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

# Segment Any 3D Object with language

## supplementary material

In this supplementary material, we provide more implementation details, introduce additional evaluation on free-form language instructions, and conduct more qualitative and quantitative analysis.

- More implementation details are provided in Sec. A.
- SOLE is evaluated on 3D visual grounding task to verify the responsive ability to free-form language instructions in Sec. B
- Analysis of CLIP visual features are provided in Sec. C.
- Variants of SOLE are introduced in terms of scalability and compatibility in Sec. D.
- Analysis of semantic-aware mask generator are provided in Sec. E.
- More qualitative results about the mask caption and segmentation results are shown in Sec. F.

## A   IMPLEMENTATION DETAILS

**Segmentation Network.** Following Mask3D (Schult et al., 2022), we use the transformer-based mask-prediction paradigm to obtain instance mask and semantic features. The masks are initialized from object queries and regressed by attention layers. For each 3D point cloud scene, we use farthest point sampling (Qi et al., 2017) to get 150 points as object queries. After getting masks from the segmentation model, we use DBSCAN (Ester et al., 1996) to break down non-contiguous masks into smaller, spatially contiguous clusters to improve the mask quality. The maximum distance and neighborhood points number are set to 0.95 and 1, respectively.

**Text Information Generation and Extraction.** To effectively generate a caption for each mask, we use a caption model in CLIP space, *i.e.*, DeCap (Li et al., 2023). DeCap is a lightweight transformer model to generate captions from CLIP image embedding. It contains a 4-layer Transformer with 4 attention heads as the language model and the visual embedding is obtained from the pre-trained ViT-L/32 CLIP model. We feed the mask features that are average pooled from the projected CLIP visual features into the DeCap model to obtain the mask caption. Then the caption is integrated into the text prompt "a {} in a scene." to better align with our data, *e.g.* "a blue chair in a scene.". With the mask caption, noun phrases are extracted by the NLP library, TextBlob (Loria et al., 2018) and spaCy (Honnibal & Montani, 2017), to get the mask-entity association.

## B   3D VISUAL GROUNDING

To further verify the effectiveness of SOLE on various language instructions, we conduct experiments on 3D visual grounding benchmark ScanRefer (Chen et al., 2020). 3D visual grounding aims at localizing 3D objects with free-form text descriptions. Therefore, we query SOLE with each text prompt in the ScanRefer validation set to get the corresponding instance, and then 3D bounding box is obtained from the instance masks. The performance is evaluated on the matching accuracy with IoU over 0.25 (ACC@25) and 0.5 (ACC@50).

**Baselines.** We compare SOLE with five specialist baseline models and one generalist model. Specialist models mean that the models are designed and trained for 3D visual grounding only while generalist models are models that can address other tasks such as instance segmentation with class names. For the specialist models, OCRand (Chen et al., 2020) uses an oracle with ground truth bounding boxes of objects, and selects a random box that matches the object category. VoteRand (Qi et al., 2019; Chen et al., 2020) leverages the pre-trained VoteNet (Qi et al., 2019) to predict bounding boxes and randomly select a box of the correct semantic class. SCRC (Hu et al., 2016) and One-stage (Yang et al., 2019b) are 2D approaches with 3D extension using back-projection. ScanRefer (Chen et al., 2020) uses a pre-trained VoteNet (Qi et al., 2019) with a trained GRU to select a matching bounding box. For the generalist model, 3D-LLM directly predicts the location of the bounding box corresponding to the text description via large language model. **Note that** except for

Table 7: **Results on ScanRefer (Chen et al., 2020) for 3D visual grounding task.** SOLE achieves the best performance on generalist models with weak supervision.

| Method | Type | Supervision | ACC@25 | ACC@50 |
|---|---|---|---|---|
| OCRand (Chen et al., 2020) | | Full | 30.0 | 29.8 |
| VoteRand (Qi et al., 2019; Chen et al., 2020) | | Full | 10.0 | 5.3 |
| SCRC (Hu et al., 2016) | Specialist | Full | 18.7 | 6.5 |
| One-stage (Yang et al., 2019b) | | Full | 20.4 | 9.0 |
| ScanRefer (Chen et al., 2020) | | Full | 41.2 | 27.4 |
| 3D-LLM (flamingo) (Hong et al., 2023) | Generalist | Full | 21.2 | — |
| **SOLE** | | **Zero-Shot** | **25.2** | **22.6** |

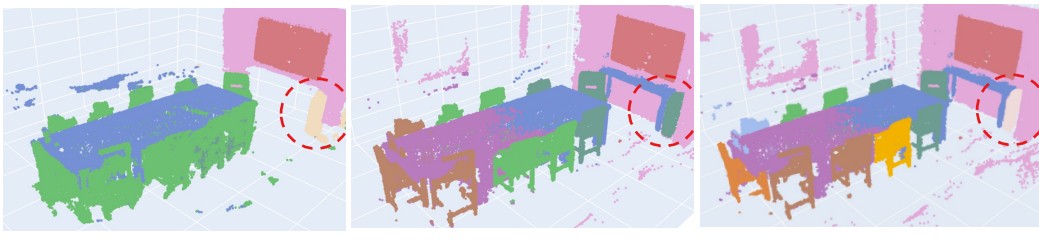

(a) K-Means clustering of $\mathbf{f}^p$     (b) K-Means clustering of $\mathbf{f}^b$     (c) K-Means clustering of $\tilde{\mathbf{f}}^b$

Figure 6: **K-means clustering of different backbone features.** Different colors denote different clusters.

OCRand and VoteRand where training is not required, the other four baseline models are trained or fine-tuned on the ScanRefer training set. Differently, SOLE is only trained with instance segmentation masks of ScanNetv2 (Dai et al., 2017), without leveraging any labels in ScanRefer.

**Results.** As shown in Tab. 7, with instance mask annotation, SOLE outperforms the generalist model 3D-LLM by 4% on ACC@25. In addition, SOLE can achieve competitive performance with the fully-supervised specialist counterpart on ACC@50 (22.6% *v.s.* 27.4%). Such results demonstrate the strong generalization ability and the effectiveness in responding to free-form language instructions of our framework.

## C ANALYSIS OF CLIP VISUAL FEATURE

CLIP visual features play an important role in the generalization ability of SOLE. In this section, we further analyze the effectiveness of CLIP visual features in the backbone feature ensemble and inference ensemble.

### C.1 BACKBONE FEATURE ENSEMBLE

As shown in Tab. 5, solely using 3D instance backbone feature $\mathbf{f}^b$ or projected CLIP visual feature $\mathbf{f}^p$ cannot achieve the best performance. 3D backbone features lack the generalized semantic information while projected CLIP visual features lack the location and geometry information. To further verify the effectiveness of the backbone feature ensemble, we visualize the clustering results of different features in Fig. 6. In Fig. 6a, all chairs are clustered together (the green cluster), showing that the projected CLIP features contain good semantic information but cannot detect the instances. In Fig. 6b, different instances within one category can be identified, *e.g.*, chairs are in three clusters. However, the semantic generalization ability is degraded. As in the highlighted red circle, the trash can is detected by projected CLIP visual features (Fig. 6a) but misclassified to a chair cluster when only using 3D backbone (Fig. 6b). Compared with using the two features separately, SOLE combines the two features and thus achieves better semantic generalization ability (segmentation of the trash can) and segmentation performance (chairs are clustered into six clusters). The visualization results further demonstrate the effectiveness of the backbone feature ensemble.

Table 8: **Analysis on classification probability ensemble.** Results are reported on the ScanNetv2 (Dai et al., 2017) dataset in 2cm voxel size.

| Component | AP | AP$_{50}$ | AP$_{25}$ |
|---|---|---|---|
| w.o. Ensemble | 42.2 | 58.6 | 66.9 |
| hard geometric mean | 43.7 | 61.1 | 70.1 |
| soft geometric mean (*ours*) | **44.4** | **62.2** | **71.4** |

Table 9: **Analysis on** $\tau$. Results are reported on the ScanNetv2 (Dai et al., 2017) dataset in 2cm voxel size.

| $\tau$ | AP |
|---|---|
| 0.1 | 44.1 |
| 0.334 | 44.2 |
| 0.667 (*ours*) | **44.4** |
| 0.9 | 42.4 |

Table 10: **Analysis of the light version of SOLE on ScanNetv2 (Dai et al., 2017).** Despite of performance drop, SOLE-*light* still outperforms mask-training methods with the fastest inference time. The best results are in **bold** while the second best results are underscored.

| Method | AP | AP$_{50}$ | AP$_{25}$ | Inference Time (s) | voxel size |
|---|---|---|---|---|---|
| OpenIns3D (Huang et al., 2023b) | - | 28.7 | 38.9 | 16.3 | 2cm |
| OpenMask3D (Takmaz et al., 2023) | 31.0 | 39.5 | 44.0 | 553.7 | 2cm |
| Open3DIS (*only 3D*) (Nguyen et al., 2024) | 31.3 | 42.4 | 47.8 | 56.6 | 2cm |
| SOLE (*ours*) | **44.4** | **62.2** | **71.4** | 454.2 | 2cm |
| SOLE-*light* | 37.1 | 50.8 | 59.0 | **13.1** | 2cm |

## C.2 INFERENCE ENSEMBLE

During inference, we combine the CLIP visual features with the predicted mask feature to achieve better generalization ability. Specifically, after obtaining the 3D masks, per-point CLIP features are pooled within the mask. The pooled CLIP feature and mask feature are then fed into the classifier to obtain the respective classification probability $p(\mathbf{f}^m)$ and $p(\mathbf{f}^p)$, and the final probability is yielded by the ensemble of them. In Tab. 8, we compare three options to ensemble the class probabilities of 3D segmentation model and CLIP model. "*w.o. Ensemble*" denotes only using the prediction of 3D segmentation model, and "*hard geometric mean*" refers to the standard geometric mean, formulated as $p(\mathbf{f}^m)^\tau \cdot p(\mathbf{f}^p)^{1-\tau}$. Our method, "*soft geometric mean*" (Eq. 7), shows the best results among the ensemble methods, demonstrating the effectiveness of dynamically fusing the prediction from two models. However, as shown in the first row, SOLE already achieves competitive performance even without utilizing the CLIP prediction, further demonstrating the strong generalization ability of our multimodal fusion network. We also analyze the parameter $\tau$ on soft geometric mean in Tab. 9. SOLE is robust to $\tau$ while achieving the best result when it is 0.667.

## D SOLE ++

As shown in Fig. 2, SOLE focuses more on the 3D semantic-aware mask generation than the semantic classification. Therefore, SOLE itself can be used for fast segmentation and it can also be combined with generalizable semantic classification and 2D mask refinement techniques proposed by previous and concurrent works (Takmaz et al., 2023; Nguyen et al., 2024). By default, we use the soft geometric mean to combine the CLIP visual feature with predicted mask feature for better generalization ability. In this section, we investigate two variants of SOLE: the light version without using any additional information for segmentation and the advanced version leveraging the semantic classification techniques from other methods.

### D.1 SOLE-LIGHT

SOLE-*light* introduces a light inference technique to exclude 2D foundation model (CLIP) used in *backbone feature ensemble* and *soft geometric mean*. In Tab. 10, we compare performance and inference time of SOLE-*light* to previous mask training methods (Takmaz et al., 2023; Huang et al., 2023b; Nguyen et al., 2024) on ScanNetv2 (Dai et al., 2017) dataset. Removing the CLIP model significantly improve the inference time by 30 times (454s *v.s.* 13s) with the performance drop to some degree. However, SOLE-*light* still outperforms previous mask training methods by a large margin, further demonstrating the strong generalization ability and good mask quality of our semantic-aware mask generator.

Table 11: **Analysis of compatibility with previous works on ScanNet200 (Rozenberszki et al., 2022).** SOLE shows high compatibility with previous mask training methods, achieving state-of-the-art performance when it is incorporated into Open3DIS framework.

| Method | 2D masks | Classification | AP | $AP_{50}$ | $AP_{25}$ |
|---|---|---|---|---|---|
| OpenMask3D (Takmaz et al., 2023) | ✗ | OpenMask3D | 15.4 | 19.9 | 23.1 |
| Open3DIS (Nguyen et al., 2024) | ✓ | Open3DIS | 23.7 | 29.4 | 32.8 |
| SOLE *(ours)* | ✗ | SOLE | 20.1 | 28.1 | 33.6 |
| SOLE + OpenMask3D | ✗ | SOLE + OpenMask3D | 20.2 | 28.5 | 33.2 |
| SOLE + Open3DIS | ✓ | SOLE + Open3DIS | **24.5** | **30.1** | **34.1** |

Table 12: **Mask quality evaluation on ScanNet200 (Rozenberszki et al., 2022).** SOLE significantly outperforms Mask3D-*CA* in terms of mask quality and the performance is even comparable to original Mask3D despite the absence of filtering step.

| Method | Semantic Guidance | filtering | AP | $AP_{50}$ | $AP_{25}$ |
|---|---|---|---|---|---|
| Mask3D-*CA* | None | ✗ | 23.1 | 29.9 | 33.4 |
| SOLE *(ours)* | MMA | ✗ | 43.5 | 56.2 | 62.5 |
| Mask3D (Schult et al., 2022) | GT labels | ✓ | 53.0 | 72.5 | 82.4 |
| SOLE *(ours)* | MMA | ✓ | 43.6 | 59.3 | 70.2 |

## D.2 SOLE-ADVANCED

Previous works (Huang et al., 2023b; Takmaz et al., 2023; Nguyen et al., 2024) commonly use the 3D masks generated by a pre-trained class-agnostic mask generator and propose techniques to refine and classify the masks with the help 2D vision large model (VLM). Since SOLE focuses more on the semantic-aware 3D mask prediction, we introduce several variants of SOLE in Tab. 11 by incorporating the previous techniques into our framework and evaluate on ScanNet200 (Rozenberszki et al., 2022). Specifically, "SOLE + OpenMask3D" further leverages the frame selection and mask feature aggregation in OpenMask3D to improve the masks classification on top of SOLE. Similarly, "SOLE + Open3DIS" uses 2D mask guidance and pointwise feature extraction in Open3DIS to improve the masks and classification. As shown in Tab. 11, our model exhibits the high compatibility with previous work and achieves the state-of-the-art performance (24.5%AP) on ScanNet200 when it is combined with Open3DIS framework.

## E  SEMANTIC-AWARE MASK GENERATOR

One of the main contributions of SOLE is that we improve the mask generator with multimodal associations and multimodal fusion network. In this section, we evaluate and analyze how semantic information improves the mask quality.

### E.1  QUALITY OF SEMANTIC-AWARE MASKS

In contrast to previous works (Huang et al., 2023b; Takmaz et al., 2023; Nguyen et al., 2024) that disregard semantic information in the mask generation, our SOLE generates semantic-aware masks with showing superior quality and generalizability. To investigate the impact of semantic information to mask quality, Tab. 12 further examines the mask evaluation in the class-agnostic setting. We first adjust Mask3D (Schult et al., 2022) to remove semantic-related parts by discarding classification head and training it on only mask annotations. This Mask3D variant, denoted as Mask3D-*CA*, is compared to SOLE in class agnostic 3DIS on ScanNet200 (Rozenberszki et al., 2022). During the evaluation, we use all of the mask proposals without filtering the low quality masks. As shown in Tab. 12, SOLE significantly outperforms Mask3D-*CA* on the mask quality, demonstrating the effectiveness of semantic information in mask generation. We also show the performance of full-supervised counterpart in Tab. 12, taking the class information into account but only evaluating the mask quality. The fully-supervised Mask3D (Schult et al., 2022) leverages classification logits to filter low-quality segments, which is their way to use semantic information to support mask generation. SOLE shows close performance to original Mask3D despite not filtering the low-quality

Table 13: **Analysis on caption influence for multimodal associations.** SceneVerse generates captions based on ground truth class labels, which can further improves the performance of SOLE.

| Method | Caption | GT Labels | AP | AP$_{50}$ | AP$_{25}$ |
|---|---|---|---|---|---|
| SOLE | DeCap | ✗ | 20.1 | 28.1 | 33.6 |
| SOLE | SceneVerse | ✓ | **26.1** | **37.3** | **44.2** |

masks. When filtering the low-quality segments, the performance of SOLE can be further improved. Comparing the second and fourth row in Tab. 12, our SOLE shows robust performance regardless of whether filtering step is present or not. Since SOLE is aware of rich semantic information, it can already generate high quality of masks without the requirements of post processing.

### E.2 QUALITY OF SEMANTIC GUIDANCE

SOLE leverages DeCap to generate mask captions without any class labels. However, although the generated captions contain rich semantic information, they are still not perfect due to the lack of class label. In Tab. 13, we replace the generated caption with captions in SceneVerse (Jia et al., 2024). SceneVerse (Jia et al., 2024) leverages LLM to generate mask captions given the mask and class labels. With the better captions, SOLE can be further improved by 6% in AP on ScanNet200 dataset. These results demonstrate that better semantic information indeed help to learn a better mask proposal modules and the label-free caption generator can still be improved, deserving investigation in the future work.

## F QUALITATIVE RESULTS

**Visualization for Segmentation Results.** In Fig. 1 and Fig. 7, we present qualitative results, demonstrating that SOLE is capable of processing free-form language queries, including but not limited to visual questions, attributes description, and functional description. More qualitative results are presented by the video in supplementary material.

**Visualization for Mask Captions.** We provide the visualization for different masks and corresponding generated captions in Fig. 8. The generated caption contains the semantic information of the 3D object as well as the location for better 3D segmentation. As shown in the examples, more than one nouns exist in the caption and thus we aggregate all the noun phrases with attention mechanism in mask-entity association.

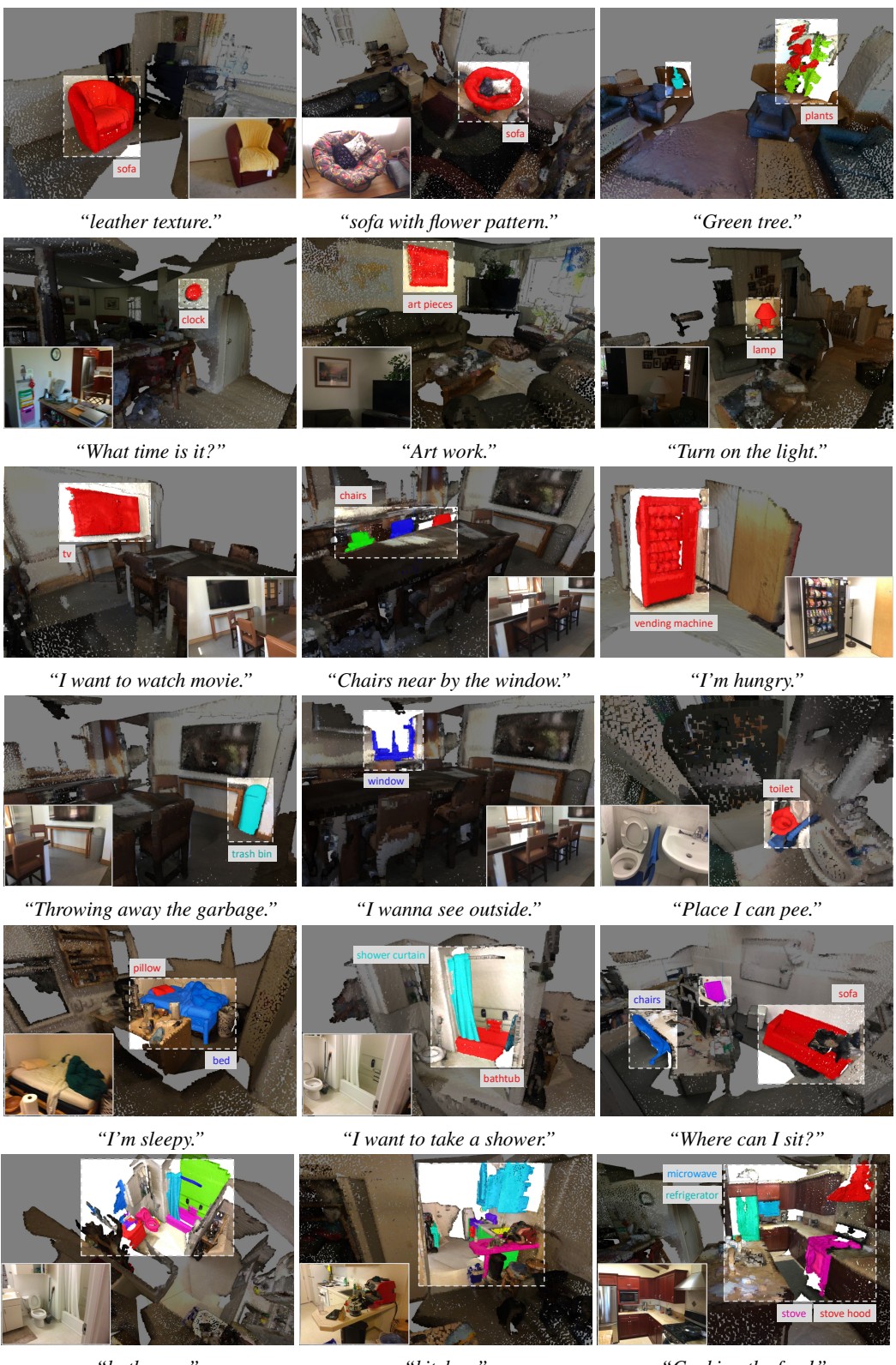

Figure 7: **Qualitative results from SOLE.** Our SOLE demonstrates open-vocabulary capability by effectively responding to free-form language queries, including visual questions, attributes description and functional description.

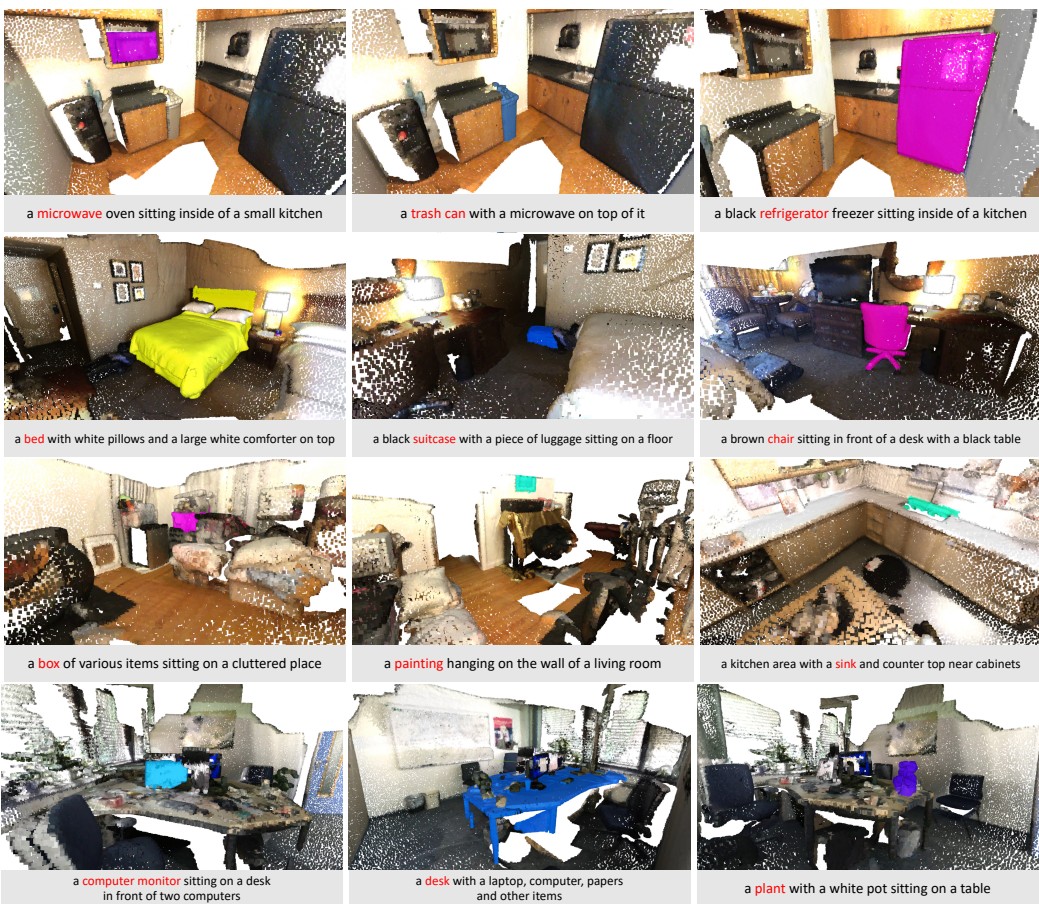

Figure 8: **Qualitative examples for mask captions.** Generated captions contain the semantic, appearance, and geometric relationship of the corresponding object.

