# OpenReview forum: "Segment Any 3D Object with Language"
_ICLR.cc/2025/Conference — ICLR 2025 Poster_

### Official Review · Reviewer_bJcr · 2024-10-31

**Soundness:** 2
**Presentation:** 3
**Contribution:** 2
**Rating:** 6
**Confidence:** 5

**Summary:**

The authors present SOLE, an approach to Open-Vocabulary 3D Instance Segmentation that uses a training-based pipeline. Basically, it proposes a new method to fuse CLIP features earlier in the network to produce more text-aware masks in testing. Particularly, to address the limitation of overfitting to seen class names in training, SOLE presents a new semantic-aware mask generation module, which integrates rich textual semantics through multimodal embeddings derived from CLIP and captioning models, allowing the model to effectively capture language-domain features. The framework introduces three multimodal association techniques (MVA, MCA, and MEA) to align 3D segmentation with language instructions, enhancing mask quality. SOLE achieves state-of-the-art performance on the ScanNetv2, ScanNet200, and Replica benchmarks.

**Strengths:**

+ Multimodal Fusion Network: SOLE’s multimodal fusion network integrates visual and language semantics, enhancing segmentation quality and generalizability.
+ Alignment with Language Instructions: The model introduces three multimodal associations to align 3D segmentation with language instructions effectively.

**Weaknesses:**

+ Still requires images in testing to extract per-point CLIP features, slowing down the whole process compared to the baselines that do not associate semantic features from text.
+ Although this paper does not use class names, they are already available, and much easier to annotate than the ground-truth 3D masks. Leveraging this information or not is up to the approaches. This cannot be claimed as an advantage.
+ Domain Adaptation Testing on ScanNet++: To assess SOLE’s generalizability, testing its domain adaptation performance on ScanNet++ [a] with over 1,500 class-agnostic categories and an open-vocabulary subset of 100 classes would be valuable. The unique point distribution in this dataset could further validate SOLE’s robustness, and strong performance here would mark a significant achievement.
+ Expanding Multimodal Associations with Large Language Models: The multimodal associations module’s versatility opens up opportunities to incorporate large language models beyond DeCap [b], including 3D-LLM [c], Ferret [d], OSPrey [e], and others.

[a] Yeshwanth, Chandan, et al. "Scannet++: A high-fidelity dataset of 3d indoor scenes." Proceedings of the IEEE/CVF International Conference on Computer Vision. 2023.

[b] Li, Wei, et al. "Decap: Decoding clip latents for zero-shot captioning via text-only training." arXiv preprint arXiv:2303.03032 (2023).

[c] Hong, Yining, et al. "3d-llm: Injecting the 3d world into large language models." Advances in Neural Information Processing Systems 36 (2023): 20482-20494.

[d] You, Haoxuan, et al. "Ferret: Refer and ground anything anywhere at any granularity." arXiv preprint arXiv:2310.07704 (2023).

[e] Yuan, Yuqian, et al. "Osprey: Pixel understanding with visual instruction tuning." Proceedings of the IEEE/CVF Conference on Computer Vision and Pattern Recognition. 2024.

**Questions:**

See Weaknesses section

---

### Official Review · Reviewer_7Fsw · 2024-11-02

**Soundness:** 2
**Presentation:** 3
**Contribution:** 3
**Rating:** 6
**Confidence:** 3

**Summary:**

This work focuses on open-vocabulary 3D instance segmentation with language instructions. To fully utilize the semantic information in the generated mask, the authors propose a framework, named Segment any 3D Object with LanguagE (SOLE). Specifically, they introduce a feature ensemble to fuse features from 3D backbone and 2D CLIP. Then, they propose a cross-modality decoder (CMD) to integrate the point-wise features and textual features. Last, the authors propose three types of association to align the mask feature with language instruction. Experimental results on ScanNetv2, Scannet200, and Replica show that the proposed method outperforms existing methods including OpenIns3D and OpenMask3D. However, I still have some questions about the methodology and experiments.

**Strengths:**

1. The proposed feature ensemble module is effective in fusing features from both the backbone and CLIP.
2. The proposed MEA effectively improves the model performance by introducing fine-grained visual-language association.
3. The proposed SOLE shows superior performance on ScanNetv2 and ScanNet200 compared to both OpenIns3D and OpenMask3D.

**Weaknesses:**

1. In Table 5, it is unclear whether the proposed CMD is effective under a voxel size of 4cm. It would be better to provide the results that remove the CMD under a voxel size of 4cm and compare with the results at the 3-th row.
2. Is the proposed method sensitive to the hyperparameters $\lambda_{MMA}, \lambda_{dice}, \lambda_{BCE}$”? More discussions are required.
3.  In Tables 1-4, it is unclear why the authors ignore the results of Open3DIS with both 2D and 3D supervision. More discussions are required.
4. In Eq(4), why not use cosine similarity to measure the distance between the mask features and the associate features? Note that using cosine similarity is common practice to measure the distance between features, and is used by OpenSeg and CLIP. Besides, on line 341, $p(\cdot, \cdot)$ should be $p(\cdot)$.
5.  As the three types of associate features lie in different feature spaces, it is difficult for a model to learn the mask features that aggregate all the advantages of associate features by Eq(4). The results in Table 6 also show that using all three types of association may not be the best choice. Why not use separate mask features for each associate feature and concatenate them together?
6. On line 457, the authors argue that “small voxel size can save the memory requirements”. In fact, using larger voxel size can reduce the number of voxels and is more memory efficient.
7. Some of the references need to be updated. For example, Open3DIS is currently published in CVPR 2024.

**Questions:**

My main concerns are about the method and experiments. Please refer to the weaknesses.

---

### Official Review · Reviewer_SCdm · 2024-11-04

**Soundness:** 3
**Presentation:** 3
**Contribution:** 3
**Rating:** 8
**Confidence:** 4

**Summary:**

This paper introduces a method for segment any 3D object with language, which tries to improve the generalizability to novel categories. The suggested method is a semantic and geometric-aware visual-language learning framework. A multimodal fusion network is given to incorporate multimodal semantics, and three types of multimodal associations as supervision are introduced to align the 3D segmentation model with various language instructions and enhance the mask quality. Overall, the suggested method outperforms previous methods.

**Strengths:**

The problem studied is interesting and has broad application prospects, particularly in the interactive understanding of 3D scenes.

This paper  is well- presentation with a clear motivation, and sufficient detail provided in the methods, making it easy to follow.

The proposed method offers a new perspective, which is to directly predict semantic-related masks from 3D point clouds with multimodal information.

**Weaknesses:**

What do the five parts in the Feature Backbone represent? Although not a contribution of this paper, it would be best to clearly illustrate this for better self-containment.

The transition from class-agnostic to class-aware is achieved by introducing point-wise CLIP features. However, converting 3D point clouds to images for point-wise CLIP feature extraction and then projecting the features back to 3D seems to have significant computational overhead.

**Questions:**

CLIP demonstrates strong generalization at the image level. How can 2D frame features be accurately transferred to point-wise features?

---

### Official Review · Reviewer_yiYW · 2024-11-06

**Soundness:** 3
**Presentation:** 3
**Contribution:** 3
**Rating:** 6
**Confidence:** 5

**Summary:**

Unlike previous methods that decouple mask proposal and open-world semantic prediction, this paper proposes the SOLE framework, which unifies these two processes into a single mask proposal framework by incorporating point-wise CLIP features into the mask proposal stage. SOLE achieves state-of-the-art results on several open-world instance segmentation benchmarks, such as ScanNetv2 and ScanNet200.

**Strengths:**

1. **[Reasonable Design]** Unlike the previous paradigm, which separates "mask proposal" and "mask understanding", this framework combines these two processes. This integrated approach seems reasonable, as the inclusion of additional features from an image foundation model is likely to improve mask proposal quality and, consequently, overall performance.

2. **[Good Results]** The reasonable design leads to improved accuracy compared to previous literature. The paper validates its performance on a sufficient number of benchmarks.

**Weaknesses:**

1. **[Potential Efficiency Issue]** A major concern relates to the original SOLE’s efficiency, particularly in terms of speed and memory consumption. Aggregating raw 2D images into a 3D point cloud is likely to be slow. Even if this process is considered a preprocessing step, the loading and processing of per-point CLIP features could also be extremely resource-intensive. This may pose a limitation for real-world applications.

2. **[Evaluation of Efficiency]** Building on the previous point, the manuscript lacks a detailed breakdown of model speed and memory consumption. This is a crucial metric for real-world applications and the potential scalability of this method. Identifying which components contribute to inefficiency would be valuable for future research. Although some numbers are provided in the appendix, this analysis is important, even if the results are not entirely favorable.

3. **[Reliance on Over-segmentation GT]** Inherited from Mask3D, the proposed method also relies on graph-based over-segmentation results, which are used for ground truth labeling in ScanNet and ScanNet200. This reliance may introduce certain issues (though originally introduced by Mask3D).

**Questions:**

For detailed questions, please refer to the weaknesses section. Regardless, I think this work is a good addition to the existing progress in 3D open-world instance segmentation. I hope to see further justification related to model efficiency.

---

### Meta-Review · Area_Chair_oVwe · 2024-12-22

**Metareview:**

This paper introduces a method named SOLE, which segments any 3D object with language, and tries to improve the generalizability to novel categories. The manuscript was reviewed by four experts in the field. The recommendations are (3 x "6: marginally above the acceptance threshold", "8: accept, good paper"). Based on the reviewers' feedback, the decision is to recommend the acceptance of the paper. The reviewers did raise some valuable concerns (especially additional and important experimental evaluations and ablation studied, needed comparisons with previous literature (clarification regarding technical insights), and together with further polishment of the manuscript) that should be addressed in the final camera-ready version of the paper. The authors are encouraged to make the necessary changes to the best of their ability.

**Additional Comments On Reviewer Discussion:**

Reviewers mainly hold concerns regarding extra additional and important experimental evaluations and ablation studies (Reviewer yiYW, 7Fsw, bJcr), needed comparisons with previous literature (Reviewer bJcr), detailed clarification on statements (Reviewer yiYW, SCdm, 7Fsw, bJcr), and together with further polishment of the manuscript (Reviewer 7Fsw, bJcr). The authors address these concerns with detailed and extra experiments and commit to polishing the revised version further.

---

### Decision · Program_Chairs · 2025-01-22

Accept (Poster)